# Collagenase-Responsive Hydrogel Loaded with GSK2606414 Nanoparticles for Periodontitis Treatment through Inhibiting Inflammation-Induced Expression of PERK of Periodontal Ligament Stem Cells

**DOI:** 10.3390/pharmaceutics15102503

**Published:** 2023-10-20

**Authors:** Yuchen Zhou, Jie Liu, Peng Xue, Jianjun Zhang

**Affiliations:** 1College of Chemical Engineering, Beijing University of Chemical Technology, Beijing 100029, China; jjlin98920@163.com (Y.Z.); 18001208450@163.com (J.L.); 2Institute of Stomatology, The First Medical Center, Chinese PLA General Hospital, Beijing 100853, China

**Keywords:** GSK2606414, nanoparticles, PERK, hydrogel, periodontitis

## Abstract

GSK2606414 is a new, effective, highly selective PERK inhibitor with adenosine-triphosphate-competitive characteristics. It can inhibit endoplasmic reticulum stress and has the possibility of treating periodontitis. However, owing to its strong hydrophobicity and side effects, highly efficient pharmaceutical formulations are urgently needed to improve the bioavailability and therapeutic efficacy of GSK2606414 in the treatment of periodontitis. Herein, a novel local GSK2606414 delivery system was developed by synthesizing GSK2606414 nanoparticles (NanoGSK) and further loading NanoGSK into a collagenase-responsive hydrogel. The drug release results showed that the drug-loaded hydrogels had outstanding enzymatic responsive drug release profiles under the local microenvironment of periodontitis. Furthermore, in vitro studies showed that the drug-loaded hydrogel exhibited good cellular uptake and did not affect the growth and proliferation of normal cells, while the drug-loaded hydrogel significantly improved the osteogenic differentiation of inflammatory cells. In the evaluations of periodontal tissue repair, the drug-loaded hydrogels showed a great effect on inflammation inhibition, as well as alveolar bone regeneration. Therefore, this work introduces a promising strategy for the clinical treatment of periodontitis.

## 1. Introduction

Periodontitis is one of the most common infectious diseases among adults. More than 30% of adults worldwide suffer from periodontitis [1,2,3]. It is also the leading cause of tooth loss. In addition, patients with periodontitis have an increased risk of diabetes [4,5], atherosclerosis [6,7], and Alzheimer’s disease [8]. Once the periodontal supporting tissue is damaged and absorbed, it is difficult to rebuild and heal by itself. At present, some methods for treating periodontal disease are applied clinically, such as guided tissue regeneration, cytokine induction, and periodontal material implantation [9,10]. These measures can only alleviate and suppress inflammation. Thus, the deconstruction of periodontal tissue cannot continue and the regeneration of periodontal tissue structure and function cannot be achieved [11,12]. In patients with deep periodontal pockets, antibiotics may be helpful as an adjunctive therapy to mechanical debridement. In addition, current treatment is ineffective for severe periodontitis. Therefore, how to effectively alleviate and inhibit the progression of periodontitis and reverse inflammation in the direction of tissue repair and reconstruction is the key problem to be solved in the treatment of periodontitis.

Inflammatory cytokines compromise the unfolded protein response (UPR) function through mortality factor (MORF)-mediated protein kinase-like endoplasmic reticulum kinase (PERK) transcription, which contributes to impaired endoplasmic reticulum (ER) function, prolonged ER stress, and defective osteogenic differentiation of periodontal ligament stem cells (PDLSCs) in periodontitis-associated chronic inflammation [13,14,15,16]. GSK2606414 is a new, potent, and highly selective PERK inhibitor. It has adenosine-triphosphate-competitive properties. It binds to the active site of PERK and inhibits the phosphorylation of eukaryotic transcription initiation factor 2α (elF2α) downstream of PERK [17,18,19]. GSK2606414 can effectively inhibit the phosphorylation of eIF2α, and thus, it can be used as an anti-inflammatory drug. However, because GSK2606414 is a small molecule with strong hydrophobicity, the oral administration route adopted in clinical research was limited by its low bioavailability and severe side effects [20,21,22,23].

Local drug delivery systems based on hydrogels are promising candidates for overcoming the drawbacks of strong hydrophobicity and high-side-effect drugs in clinical treatments. In particular, smart hydrogels with stimuli-responsive characteristics have promising advantages in drug delivery, including on-demand release, long action time, and low administration frequency, which lead to significantly improved drug bioavailability and side effects [24]. Among smart hydrogels, enzymatic-responsive hydrogels have attracted extensive attention due to their high sensitivity and specificity [25]. In periodontal tissue, when there is dental plaque stimulation and periodontal disease, the expression and activity of matrix metalloproteinases (MMPs) are pathologically increased [26], especially MMP-2, which is a key enzyme involved in connective tissue degradation and tissue remodeling that is overexpressed in gingival crevicular fluid, gingival tissue, and saliva samples from patients with chronic periodontitis [27]. Therefore, hydrogels with MMP-2 responsiveness would be an ideal carrier for local drug delivery in periodontal treatment aiming at achieving optimal therapeutic efficacy while reducing the side effects of drugs.

When using smart hydrogels as local drug carriers, the drugs need to be homogeneously distributed in the hydrogel network to achieve a constant release rate through stimuli factors. However, water-insoluble drugs are difficult to stably disperse in hydrogel matrices [28]. One effective method to address this problem is preparing drug nanoparticle aqueous suspensions [29]. The high-gravity nanoprecipitation technique (HGNPT) in a rotating packed bed (RPB) was developed as an effective method for drug nanoparticle preparation [30]. Owing to intensified mass transfer and micromixing efficiency between the drug solution and antisolvent under high-gravity conditions, drug nanoparticles with smaller particle sizes, more uniform particle size distribution, and high aqueous dispersibility were perfectly obtained [31].

In this work, we developed a local GSK2606414 delivery system based on collagenase-responsive hydrogels for periodontitis treatment by inhibiting the inflammation-induced expression of PERK in periodontal ligament stem cells. As shown in Figure 1, we first prepared GSK2606414 nanoparticles (NanoGSK) via HGNPT with amphiphilic PDLLA-PEG-PDLLA copolymers as drug excipients. Then, NanoGSK was encapsulated into a collagenase-responsive hyaluronic acid (HA) hydrogel formed using the Michael addition reaction between acrylated HA (HA-AC) and an MMP-2-cleaved peptide cross-linker (MMPc). For the in vivo evaluation of the periodontitis treatment, the mixed solution of HA-AC and NanoGSK was directly injected with the MMPc solution at the gingival sulcus via a disposal-connected mixing system, and a drug-loaded hydrogel depot was formed in situ. Finally, due to the overexpression of MMP-2 in the gingival tissue of periodontitis, the hydrogel depot was continuously degraded to release the NanoGSK cargo, and NanoGSK further spread and entered the PDLSCs to inhibit the inflammation-induced expression of PERK. Moreover, in vitro and in vivo studies demonstrated that NanoGSK-loaded hydrogels exhibited outstanding inflammation inhibition effects and alveolar bone repair ability [32,33].

## 2. Materials and Methods

### 2.1. Materials

Polyethylene glycol, stannous octoate, and collagenase were purchased from Sigma-Aldrich (St. Louis, MO, USA). Dichloromethane and n-hexane were purchased from Shanghai Macklin Biochemical Co., Ltd. (Shanghai, China). Deuterated chloroform was obtained from Bailingwei (Beijing, China). D, L-lactide was purchased from Jinan Daigang Biological Engineering Co., Ltd. (Beijing, China). GSK2606414 was purchased from GLPBIO (Montclair, CA, USA). FITC was acquired from Aladdin (Beijing, China). Lipopolysaccharides derived from *E. coli* 055:B5 were obtained from Sigma-Aldrich. α-modified Eagle’s medium (α-MEM), fetal bovine serum, and penicillin/streptomycin were provided by Gibco (San Diego, CA, USA). PDLSCs were obtained from the Institute of Dentistry, First Medical Center, General Hospital of the Chinese People’s Liberation Army.

### 2.2. Synthesis of PDLLA-PEG-PDLLA

To a round-bottom flask, 60 g polyethene glycol, 10 g D-lactide, and 10 g L-lactide were added, and the water was removed from the medicine by vacuuming and dissolving it at 110 °C. The argon system was turned on to add fully inert gas into the reaction system, and 24 mg of stannous octoate was added under the protection of inert gas. After the reaction was initiated, the system was heated to 130 °C. After 18 h of reaction, heating was stopped and the system was allowed to cool completely. Purification was achieved by repeatedly dissolving in dichloromethane and settling in n-hexane, where the volume ratio of dichloromethane to n-hexane was 1:5. The purified product was dried to obtain the product PDLLA-PEG-PDLLA, which was stored at −20 °C for later use.

### 2.3. Preparation and Characterization of NanoGSK

In the high-gravity rotating packed bed experiment, the supergravity optional packed bed was combined with 200 mL of 1% PDLLA-PEG-PDLLA PBS buffer solution (150 mM, pH = 7.4) and 400 mg of GSK2606414 in 10 mL DMSO. The first product was obtained following a predetermined reaction time. For 8 h, the product was dialyzed in ultrapure water. A dialysis bag with a 3500 Da molecular weight cutoff was used. Following dialysis, the product was purified by removing the free GSK2606414 and solvent DMSO. NanoGSK drug powder was then freeze dried and kept at −20 °C.

An appropriate amount of NanoGSK suspension after dialysis was taken and diluted with an appropriate amount of ultrapure water, and its particle size and potential were characterized using a dynamic light scattering particle size analyzer (DLS). The sample suspension was dropped onto silicon wafers and copper grids for scanning electron microscopy (SEM) and transmission electron microscopy (TEM) analysis.

The volume of the drug nanoparticle suspension was collected and measured after the dialysis. After redispersing it evenly, 100 μL was placed into a centrifuge tube, freeze dried, and weighed; then, it was dissolved in 2 mL of DMSO. After dissolution, the absorbance at the maximum absorption wavelength was measured using an ultraviolet-visible spectrophotometer, and the drug loading (LC) and encapsulation efficiency (EE) of the nanomedicine were calculated according to the standard curve at the maximum absorption wavelength. The formulas are as follows:(1)$$\mathrm{LC}\;\left(\mathrm{wt}\%\right)=\frac{\left(\mathrm{weight}\;\mathrm{of}\;\mathrm{loaded}\;\mathrm{drug}\right)}{\left(\mathrm{weight}\;\mathrm{of}\;\mathrm{drug}\;\mathrm{loaded}\;\mathrm{nanocapaules}\right)}\times 100\%$$
(2)$$\mathrm{EE}\;\left(\mathrm{wt}\%\right)=\frac{\mathrm{weight}\;\mathrm{of}\;\mathrm{loadeed}\;\mathrm{drug}}{\mathrm{weight}\;\mathrm{of}\;\mathrm{initially}\;\mathrm{added}\;\mathrm{drug}}\times 100\%$$

### 2.4. Preparation and Characterization of HA-AC

In 400 mL ultrapure water, 3.0 g HA (40 kDa) was dissolved; 37.5 g adipic acid dihydrazide (ADH) was added after the HA had completely dissolved, and the pH was adjusted to 4.75. Then, 6 g 1-ethyl-3-[3-dimethylaminopropyl] carbodiimide hydrochloride (EDC) was added, and the pH was adjusted to 4.75 until it was stable and reacted for 12 h. The reacted solution was purified in ultrapure water with a dialysis bag with a molecular weight cutoff (MWCO) of 8000 Da for two days, during which time the water was changed 4 times a day, and the product HA-ADH was obtained by freeze drying after dialysis. Two grams of HA-ADH was dissolved in a 400 mL HEPES buffer solution (pH = 7.2), and then 1.4 g N-acryloxysuccinimide (NAS) was dissolved in 10 mL DMSO while being slowly added dropwise to the above solution and reacted for 12 h. The reacted solution was purified in ultrapure water using a dialysis bag with a molecular weight cutoff (MWCO) of 8000 Da for two days, and the product HA-AC was obtained by freeze-drying after dialysis. The structure of the product HA-AC was confirmed via ^1^H NMR using deuterated water (D_2_O) as the solvent.

### 2.5. Gelation and In Vitro Degradation of the HM Hydrogel

HA-AC (12 mg) was dissolved in 160 μL triethanolamine (TEOA) buffer (300 mM, pH = 8), and 2 mg HS-MMP-SH cross-linker (the MMPc peptide sequence was GCRD GPQG IWGQ DRCG) was dissolved in 40 μL TEOA buffer. The above two solutions were mixed and placed in a thermostat at 37 °C to obtain the HA-MMPc hydrogel (HM) after gelation.

The in vitro degradation of HM (6% HA-AC) was incubated in PBS (150 mM, pH = 7.4) with or without collagenase (5 U/mL) at 37 °C. A simple PBS buffer was used as the control group, and the experimental group was used as the PBS buffer containing 5 U/mL collagenase. At specific time points (0, 1, 3, 5, 7, 10 days), the hydrogel was removed, and the surface liquid was blotted to accurately weigh the remaining hydrogel mass.

### 2.6. Drug Loading and Drug Release of the HM Hydrogel

NanoGSK (40 mg) was accurately weighed and dispersed in 160 μL of TEOA buffer (300 mM, pH = 8.0) ultrasonically. Then, 12 mg HA-AC was thoroughly mixed with the above solution until the HA-AC was completely dissolved. In 40 μL TEOA buffer solution, 2 mg MMPc agent was dissolved. The above two solutions were mixed and put in a 37 °C incubator. After gelation, NanoGSK-loaded HA-MMPc hydrogel (NGHM) was prepared. The hydrogel was expanded in PBS buffer for 2 h, removed and freeze dried in a freeze-drying machine, dipped in liquid nitrogen, and broken apart. The hydrogel was fixed on the vertical section sample table, and the section shape was observed using SEM after gold spraying.

PBS buffer solution (150 mM, pH = 7.4) was used to simulate the human environment to research drug release. A dialysis bag with a molecular weight cutoff of 3500 Da and a diameter of 28 mm was prepared. One end was sealed with a thin thread and FreeGSK, NanoGSK, NGHM, NGHM + collagenase (5 U/mL) with the same drug content, and 2 mL PBS buffer solution were added. After that, the other end of the dialysis bag was sealed with a thin thread, put into a 50 mL screw-top glass serum bottle, and 20 mL PBS buffer solution was added. Three groups were prepared in parallel. At a set point in time, 2 mL of the sample was removed from the bottle, 2 mL of fresh PBS buffer solution (150 mM, pH = 7.4) was added, and 5 U/mL PBS buffer solution was added to the fourth group. The released GSK2606414 concentrations in the 2 mL solutions were measured using a UV-vis spectrophotometer at the maximum absorption. Then, the cumulative release of the drug (CR) was calculated according to the following formula:(3)$$\mathrm{CR}\left(\mathrm{wt}\%\right)=\frac{\mathrm{The}\;\mathrm{cumulative}\;\mathrm{release}\;\mathrm{of}\;\mathrm{the}\;\mathrm{drug}\;\mathrm{at}\;\mathrm{this}\;\mathrm{time}}{\mathrm{Total}\;\mathrm{amount}\;\mathrm{of}\;\mathrm{drug}}\times 100\%$$

### 2.7. In Vitro Uptake Experiment

Since GSK2606414 is not fluorescent, only FITC-labeled GSK2606414 was added to the cell uptake assay. Live–dead assays and cytotoxicity assays were performed using GSK2606414 that was not labeled with FITC.

The PDLSCs cultured to the 2nd to 5th passages were digested, a cell suspension of 5 × 10^4^ cells/mL was prepared, and the suspension was inoculated in an 8-well plate. A total of 200 μL of the cell suspension was added to each well, and we waited for the cells to adhere to the wall. After this, the control group (NS) and the experimental group were set up, namely, NanoGSK-FITC and NGHM-FITC (10 μg/mL), respectively, and cultured in a cell incubator for 6 h. The drug and medium were aspirated and washed three times with sterile PBS buffer. Cells were fixed by adding 100 μL of paraformaldehyde for 10 min. After removing the paraformaldehyde, the cells were washed 3 times with sterile PBS buffer, stained with DAPI (1 μg/mL) for 2~5 min, and then washed 3 times with PBS. To observe the cell uptake by CLSM, the FITC excitation wavelength was 490 nm, the emission wavelength was 494/525 nm, and the DAPI excitation/emission wavelength was 405 nm/460 nm.

### 2.8. In Vitro Cytotoxicity

PDLSCs that have been grown through the 2nd to 5th passages should be digested before being inoculated in a 96-well plate with 100 μL of the cell suspension in each well. The original liquid was aspirated, the cells were washed three times with sterile PBS buffer solution, 100 μL of fresh complete medium was added, and 20 μL of NanoGSK was added at a series of concentrations after the cells had adhered to the wall. Each group had three replicate wells, and the real drug solubilities were 0, 5, 10, 20, 50, and 100 μg/mL. The incubation period was set to 12 or 24 h. After incubation for a fixed time, the drug-containing medium was aspirated, 100 μL of fresh medium was added, 20 μL of MTT solution (5 mg/mL) was added, and the mixture was shaken gently. The original medium was aspirated and washed three times after incubation for 4 h. The blue-purple crystalline formaldehyde produced by living cells was dissolved in 100 μL of biograde DMSO and incubated for 5 min in a cell incubator. Consider using an enzyme-linked immunoassay. The OD values at 570 and 630 nm were measured, and then the following calculation was used to determine the cell survival rate:(4)$$\mathrm{Cell}\;\mathrm{viability}=\frac{A_{\mathrm{sample}}}{A_{\mathrm{control}}}\times 100\%$$

The PDLSCs cultured to the 2nd to 5th generations were digested, and a cell suspension of 5 × 10^4^ cells/mL was prepared. The suspension was inoculated in an 8-well plate, 200 μL of the cell suspension was added to each well, and the cells were allowed to adhere to the wall. After that, the control group (NS) and three experimental groups were set up, namely, FreeGSK, NanoGSK, and NGHM (50 μg/mL), and cultured in a cell incubator for 24 h. The medicine and culture media were aspirated and rinsed with sterile PBS buffer solution 3 times. To prepare the staining working solution, we dissolved 5 μL Calcein AM and 15 μL PI in a 5 mL 1 × Assay Buffer. Then, 100 μL of staining working solution was added to each well, incubated in a cell incubator for 15–30 min, and washed with sterile PBS buffer solution 3 times. The statuses of live and dead cells were observed using CLSM. Live cells show yellow-green fluorescence with excitation/emission wavelengths of 490 nm/517 nm; dead cells show red fluorescence with excitation/emission wavelengths of 545 nm/617 nm.

### 2.9. Cell Proliferation Experiment

According to the methodology described in Section 2.8, the cells were inoculated in a 96-well plate. After the cells adhered to the wall, the original medium was aspirated, the cells were washed 3 times with sterile PBS buffer solution, and 100 μL of fresh complete medium was added. The control group and the experimental groups NS, FreeGSK, NanoGSK, blank Gel, and NGHM were set; the volume of each group was 20 μL; the drug concentration of each group was the final concentration of 50 μg/mL; and the cells were cultured for 1, 2, 4, and 7 days. The same MTT method was used to calculate the cell proliferation rate, which was then calculated using the original value of the number of adherent cells added.

### 2.10. In Vitro Osteogenic Capacity

The 2nd-to-5th-generation PDLSCs were inoculated in a 6-well plate at a cell density of 1 × 10^5^ cells/mL. A control group and an experimental group were set up. To imitate the cellular milieu of inflammation in periodontal disease, the experimental group was given LPS (10 μg/mL) to promote inflammation. After inflammation was induced, the osteogenic induction fluid of different experimental groups was replaced (take 100 mL cell culture medium as an example, which contains 0.306 g β-glycerophosphate, 20 μL dexamethasone, and 0.167 mL vitamin C, the preparation process should be protected from light as much as possible and stored in the refrigerator at 4 °C). The experimental groups were the control, LPS, LPS + blank Gel, LPS + FreeGSK (50 μg/mL), LPS + NanoGSK (50 μg/mL), and LPS + NGHM (50 μg/mL). We used Western blotting to detect the expressions of the osteogenic-related proteins Runx2 and OCN, and quantitative reverse transcription polymerase chain reaction (qRT-PCR) to detect the relative mRNA expressions of the osteogenic-related genes Runx2 and OCN, and the primers for qRT-PCR were showed in Table 1. After 7 days of osteogenic induction, RNA was extracted using a centrifugal column method, and the relative expressions of osteogenic-related mRNAs were detected using the 2^−∆∆Ct^ method (Ct is the number of cycles). After 14 days of osteogenic induction, the cells were lysed to obtain protein solutions, and protein extracts (20 μg per sample) were separated using sodium dodecyl sulfate-polyacrylamide gel electrophoresis and then electroblotted onto polyvinylidene difluoride membranes. After sealing for 2 h, the membranes were reacted with horseradish-peroxidase-conjugated goat anti-rabbit IgG. Finally, we used chemiluminescence to observe the bands.

After alizarin red staining, the same cells were cultivated and subjected to osteogenic induction for 28 days. The cells were then removed from the media, fixed in 95% ethanol for 10 min, washed three times in PBS, stained in 1% Alizarin Red staining solution for 30 min in a cell incubator, and then rinsed five times in PBS. Subsequently, photographs were taken using an inverted microscope. Then, the mineralized nodules were dissolved in CPC solution, and the absorbance at 562 nm was detected using an enzyme marker.

### 2.11. Periodontal Tissue Regeneration In Situ

LPS injection was used to establish a Sprague Dawley (hereinafter referred to as SD) rat periodontitis model. Considering that the rat’s incisor teeth can always grow, it is not suitable as an experimental object. The rat’s molars were used as the experimental teeth. Thirty six-week-old Sprague Dawley rats (weight 180 ± 20 g) were provided by the Beijing Weitong Lihua Laboratory Animal Technology Co., Ltd. (Beijing, China). Each animal had at least a 7-day transition period prior to surgery. All the animal protocols in this study were conducted with the approval of the Animal Study Committee of Chinese PLA General Hospital (no. 2021-X17-91). All experiments were performed in compliance with the Chinese PLA General Hospital’s policy on animal use and ethics. The rats were randomly divided into five groups (*n* = 6/group). The modeling process was as follows: the rats were weighed and anesthetized via an intraperitoneal injection of 1% sodium pentobarbital solution (dose 40 mg/kg body weight). When smooth breathing, muscle relaxation, skin pain, and eyelid irritation disappeared, the rat’s anesthesia was successful. The rat was fixed on the experimental table, the head was tilted back as much as possible, and the mouth was pulled open to fully expose the maxillary molars and gum tissue. The mucosa around the selected experimental teeth were disinfected and evenly smeared with 1% iodine tincture. The palatal gingiva between the maxillary first molar and the second molar was used as the injection point. The experimental groups were LPS (10 μg/mL), LPS + blank Gel, LPS + FreeGSK (50 μg/mL), and LPS + NGHM (50 μg/mL). The rats were injected every other day for a total of 3 times, and samples were collected 21 days after the first injection. The state of each rat was observed using the naked eye, and its gum, periodontal, and oral hygiene were recorded. A micro-CT scan was performed to trim the upper jaws of the SD rats to a suitable size, and the scans and reconstructions were performed on the micro-CT machine. Following the micro-CT scans, samples were decalcified in 10% ethylene diamine tetraacetic acid (EDTA) for three months, with the EDTA solution replaced every three days. After decalcification, the samples were dehydrated in an ethanol gradient and embedded in paraffin. The samples were stained with H&E to observe the degree of alveolar bone loss and inflammatory response. To assess the collagen fibrils, Masson staining was performed on the samples. The positive area of collagen fibers was calculated using ImageJ 1.53p software (National Institutes of Health, Bethesda, MD, USA). Tissue sections were subjected to immunohistochemistry (IHC) staining to detect the PERK expression levels.

### 2.12. Statistical Analysis

All data were presented as the mean ± standard deviation (SD) of at least three independent experiments. All statistical analyses were performed using GraphPad (InStat, La Jolla, CA, USA). Significant intergroup differences were determined using one-way ANOVA analysis. A difference was considered to be statistically significant if the probability value was less than 0.05 (*p* < 0.05).

## 3. Results and Discussion

### 3.1. Preparation and Characterization of NanoGSK

NanoGSK was prepared using HGNPT. PDLLA-PEG-PDLLA self-assembled to form a hydrophilic shell and a hydrophobic core to form nanocellular bundles. The formation of the hydrophobic core provided a stable loading space for the hydrophobic drug GSK2606414. The drug nanoparticle could be uniformly dispersed in the aqueous phase (Figure 1a). The synthetic route of PDLLA-PEG-PDLLA is shown in Appendix A. To confirm the successful synthesis of the polymer, nuclear magnetic resonance hydrogen spectroscopy experiments were performed. Through the ring-opening reaction, PEG polymerized with D/LLA to form the amphiphilic polymer PDLLA-PEG-PDLLA. As shown in Appendix A, the polymer had a proton peak at 3.7 ppm which belong to methylene of the PEG block, and the peaks at 5.2 ppm and 1.8 ppm contribute to methine and methyl of the PDLLA block, which indicates the successful synthesis of PDLLA-PEG-PDLLA. The particle size of NanoGSK was measured using a dynamic light scattering particle size meter to be 342 nm with a narrow particle size distribution, indicating that the prepared nanomedicine was homogeneous in size and stably dispersed (Figure 1a). Moreover, the particle size distribution was narrow, indicating that the prepared drug nanoparticle had a uniform particle size. Meanwhile, the zeta potential of PDLLA-PEG-PDLLA was measured to be −3.92 mV (Appendix A), the zeta potential of the same concentration of NanoGSK was −8.83 mV, and the absolute value of the nanosized drug potential was greater, thus indicating that our prepared nanoparticles were more stable. As observed under scanning electron microscopy (Figure 1b) and transmission electron microscopy, NanoGSK was spherical. Under transmission electron microscopy (Figure 1c), the core–shell structure of the nanomedicine can be clearly seen and had a good coating effect on the drug.

### 3.2. Preparation and Characterization of Drug-Loaded Hydrogel

HA-AC was synthesized by HA with ADH followed by HA-ADH modification with NAS under the activation of EDC, as shown in Appendix A. The ^1^H NMR spectra showed typical methylene peaks at 1.6 ppm and 2.3 ppm and typical vinylene peaks at 6.4 ppm, which indicate that ADH and NHS were successfully grafted onto HA (Appendix A).

As the -C=C- in the side chain of HA-AC and the -SH at the ends of the 16-peptide cross-linker MMPc can undergo Michael addition, the two can be dissolved in TEOA buffer solutions and mixed under mild conditions. During the reaction, the long-chain molecules cross-link to form a three-dimensional network structure, resulting in a hydrogel. The structure of this hydrogel is a three-dimensional network structure with internal pores reaching several tens of microns. It can therefore be used to load a variety of particles with large spaces. NGHM was made by adding NanoGSK prior to gelation (Figure 1b). The hydrogel had a fixed shape and a degree of adhesion (Figure 2a). A cross-sectional view of the hydrogel obtained using SEM shows the NanoGSK attached to the pore walls of the NGHM hydrogel (Figure 2b,c).

The rate of degradation of the prepared HM hydrogels was significantly different when placed in PBS-buffered solutions with or without collagenase (Figure 2d). In the PBS-buffered solution without collagenase, the HM hydrogels were completely degraded within approximately 18 days. This indicates that in normal humans, HM hydrogels can be completely degraded without residue and have little effect on the human body. In a PBS-buffered solution containing 5 U/mL collagenase, the degradation of the HM hydrogels was even faster because the polypeptide chains in the cross-linker MMPc were selectively cut in the presence of collagenase. The disruption of the three-dimensional network structure of the hydrogel accelerated the rate of degradation of HM.

To investigate the in vitro release behavior of the drug, the enzymatic response properties of HM drug-loaded hydrogels were verified by simulating the periodontitis environment. The drug-loaded hydrogels were exposed to a PBS-buffered solution containing 5 U/mL collagenase to simulate the microenvironment of periodontitis. FreeGSK, NanoGSK, and NGHM were placed in a PBS-buffered solution at 37 °C, and the drug release behavior was studied and compared. As shown in Figure 2e, FreeGSK showed a very pronounced dissolution behavior in the early stages. After 12 h, the drug release rate reached 30% and could not continue increasing. The reason for this was that this small molecule inhibitor has a very low solubility and does not dissolve completely in aqueous solutions, which also limits its absorption in humans.

NanoGSK is a nanosized version of a hydrophobic drug. Due to the small size of nanosized drug particles, they have a significant burst release behavior in PBS solutions [34]. Within 24 h, the cumulative release of the drug reached 70%. In subsequent trials, the cumulative release of the drug reached 80%, which considerably increased the solubility of the drug [35]. This indicates that the problem of difficult absorption caused by the hydrophobicity of the drug was effectively addressed. The drug-loaded hydrogel NGHM had some burst release behavior in the early stages [36]. This was because during the preparation of the drug-loaded hydrogel, part of the NanoGSK was attached to the outer surface of the hydrogel, and this part of NanoGSK could be rapidly dispersed in the PBS-buffered solution so that the cumulative release of the drug reached 20% after 24 h [37]. This abrupt release behavior can greatly alleviate the initial inflammatory symptoms during the treatment of periodontitis [24]. In the subsequent drug release behavior, NGHM was released slowly in a collagenase-free environment, reaching 42% after 21 days, showing a good drug release effect. In the presence of 5 U/mL collagenase, NGHM could be released more rapidly and consistently. A 71% cumulative drug release could be achieved after 21 days, 1.7 times that of the collagenase-free environment. This marked increase in drug release demonstrates that NGHM had a significant collagenase-responsive drug release behivor.

### 3.3. In Vitro Cellular Uptake and Cell Cytotoxicity of NGHM

The cellular uptake was observed with CLSM, where the nuclei that were stained with DAPI staining showed blue fluorescence under excitation light and the FITC-modified NanoGSK showed green fluorescence under excitation light. As shown in Figure 3a,b, the nuclei of the control cells showed a blue oval shape. In the NanoGSK and NGHM groups, both blue fluorescence and green fluorescence were evident and superimposed to show that most of the cells had taken up NanoGSK. This indicates that the nanoscale drug can be taken up by the cells, thus improving the bioavailability of the hydrophobic drug GSK2606414.

After 24 h of cell culture, cells were stained with the live–dead kit and visualized as live or dead with CLSM. Green fluorescence under excitation light was imaged for live cells, and red fluorescence was imaged for dead cells. As shown in Figure 3c, in the blank control group, the cells all showed green fluorescence, indicating a good cell state. In the FreeGSK-treated group, there were a few dead cells that presented red fluorescence. Furthermore, both the NanoGSK- and NGHM-treated groups showed that most of the cells presented green fluorescence, suggesting that NanoGSK and NGHM had a low cytotoxicity effect on the PDLSCs.

The MTT method was used to quantitatively analyze the effect of different concentrations of NanoGSK on the survival rate of PDLSCs, and the culture times were 12 h and 24 h. It can be seen from Figure 3c that NanoGSK was less toxic to cells. At higher concentrations of NanoGSK, the cell survival rate reached 90% after 24 h of incubation at 100 μg/mL. The highest cell survival rate was achieved at a concentration of 50 μg/mL, and this concentration was used for subsequent experiments. From the results of the cell proliferation experiments of PDLSCs shown in Figure 3d, it can be seen that in the control group, the cells showed good growth with increasing culture time and could reach the original cell number after 7 days of culture. The addition of the blank hydrogel had no significant effect on cell growth, indicating that the hydrogel had good biocompatibility as a drug carrier. In the experimental group where the drug was added, the hydrophobicity of FreeGSK affected the cell proliferation to a certain extent. The cultures with NanoGSK and NGHM reached more than 8 times the original cell number after 7 days of culture. There was a pro-proliferative effect on the cell growth compared with the control group. Therefore, these results suggest that NGHM promotes the proliferation and uptake of PDLSCs better than FreeGSK and NanoGSK.

### 3.4. In Vitro Expression of Osteogenic Markers and Mineralized Nodule Formation of NGHM

Inflammation was induced in healthy PDLSCs with LPS (10 μg/mL) to mimic the periodontitis microenvironment. After the induction of inflammation, the corresponding samples were added according to the experimental group settings and observed after coculture, as shown in Appendix A. Compared with the control group, the number of cells decreased after the induction of inflammation and increased after the addition of the drug, reflecting the effect of the drug on inflammation, which was the most significant in the NGHM group with good cell status.

To detect the difference in osteogenic differentiation between healthy PDLSCs and PDLSCs under inflammatory conditions in each experimental group, the expression levels of the associated osteogenic proteins Runx2 and OCN were detected using Western blotting after the induction of osteogenic differentiation. As shown in Figure 4a, the expression levels of the cell-associated osteogenic proteins Runx2 and OCN were lower than those in the control group. The osteogenic expression ability of cells did not change after the addition of the blank hydrogel. The results indicate that the hydrogel material had no effect on the osteogenic differentiation ability. The osteogenic differentiation ability was increased after the addition of the drug. The osteogenic differentiation ability of the cells was gradually enhanced after the addition of FreeGSK, NanoGSK, and NGHM. This indicates that the bioavailability of GSK2606414 was increased and the toxicity was decreased. NGHM had the highest osteogenic differentiation capacity. This was due to the controlled and slow release of the hydrogel. A controlled release allows for the release of the drug under inflammatory conditions. This effect enables the drug to be released continuously so that the cells are at a relatively stable drug concentration and the absorption and utilization of the drug are more effective at achieving a good therapeutic effect. Its osteogenic differentiation ability is close to that of normal cells.

After the cells differentiate into osteoblasts, there will be calcium deposits, known as calcium nodules, on the cell surface. Alizarin red can react with calcium to produce a deep red substance, the color of which can indicate the role of bone formation. As shown in Figure 4b, the LPS group and the LPS + blank Gel group induced by osteogenic differentiation formed fewer calcium nodules and were brighter in color than those formed via the osteogenic differentiation of normal PDLSCs. The number of calcium nodules in cells under inflammatory conditions with the addition of FreeGSK, NanoGSK, and NGHM gradually increased, and the color gradually became darker. These results indicate that the cells under inflammatory conditions with the addition of blank hydrogel had a weaker osteogenic differentiation ability, and the cells with the addition of FreeGSK, NanoGSK, and NGHM had a progressively stronger osteogenic differentiation ability. The results of the CPC quantitative analysis were similar (Figure 4c). The results were related to osteogenic-differentiation-related proteins and genes. The expression test results were consistent. After 7 days of osteogenic induction, qRT-PCR was employed to detect the expression of the osteogenic genes Runx2 and OCN. The relative intensity indicated the gene expression of each group using the LPS group as a control, and values were calculated according to the 2^−∆∆Ct^ method. As shown in Figure 4d,e, the relative expressions of the mRNA levels of the Runx2 and OCN genes related to the osteogenic induction group of PDLSCs decreased after LPS stimulation, and the osteogenic differentiation ability was diminished. The FreeGSK, NanoGSK, and NGHM osteogenic induction groups gradually increased the osteogenic mRNA expression, indicating that the NGHM hydrogel could improve the osteogenic differentiation ability of the cells because NanoGSK use significantly restored the osteogenic differentiation of PDLSCs in the inflammatory microenvironment.

### 3.5. The Hydrogel Delivery System Decreased Inflammation and Promoted Periodontal Bone Tissue Regeneration In Vivo

From the macroscopic observation, the sanitary conditions in the oral injection area of SD rats in the LPS group and LPS + blank Gel group were poor: food and soft dirt accumulated at the edge of the gums, there was congestion and swelling in the gum tissue, and periodontal tissue erosion and atrophy were severe in the operation area. The oral conditions of SD rats treated with LPS + FreeGSK (50 μg/mL) and LPS + NGHM (50 μg/mL) gradually improved. Among them, the effect was obvious in the NGHM group. The erosion and atrophy of the tissue improved significantly, and the gum tissue had slight congestion and swelling.

Micro-CT images of normal rat teeth are shown in Figure 5a. The alveolar bone of the rats injected with LPS was destroyed (Figure 5b), with obvious bone resorption, and the periodontitis model was successfully established. Compared with the LPS group, in the LPS + blank Gel group, as shown in Figure 5c, the addition of blank hydrogel had little effect on the teeth and did not further expand the bone resorption. As shown in Figure 5d, in the LPS + FreeGSK (50 μg/mL) group, the volume of alveolar bone in the furcation area of the tooth root increased, indicating that FreeGSK had a certain effect on the treatment of periodontitis. As shown in Figure 5e, the LPS + NGHM (50 μg/mL) group bone resorption was significantly reduced, and alveolar bone reconstruction could be completed under the condition of inducing inflammation, indicating the good therapeutic effect of NGHM. Figure 5f shows the distance between the cementoenamel junction (CEJ) and the alveolar bone crest (ABC). Compared with the LPS group, the CEJ–ABC distance of the LPS + NGHM group decreased, where it was similar to the sham group, suggesting that NGHM can significantly repair the loss of alveolar bone in rats with periodontitis.

To further research the regeneration of periodontal tissue, H&E staining was performed to evaluate the inflammation status and periodontal tissue regeneration between the maxillary first and second molars. The normal periodontal tissue structure in the sham group is shown in Figure 6. The morphology of periodontal tissue was complete, the periodontal fibers were arranged in an orderly manner, and there was no massive inflammatory cell infiltration. Nevertheless, periodontal ligament fibers were placed in a disorderly manner, and there was considerable inflammatory cell infiltration in the LPS group. In addition, the LPS group also showed a greater distance between the cementoenamel junction (CEJ) and the alveolar bone crest (ABC) than the sham group. However, supplementation with NGHM inhibited periodontal destruction, showing a CEJ–ABC distance similar to that of the sham group. Furthermore, the arrangement of the periodontal ligament fibers in the NGHM hydrogel group tended to be normal, and the infiltration of inflammatory cells was lower than that in the LPS group. It is significant that the NGHM hydrogel has a strong ability to regenerate periodontal tissue.

Histological sections stained with Masson’s trichrome are shown in Figure 7a. The periodontitis-diseased rats experienced severe collagen loss. The sham group and NGHM hydrogel group had more abundant blue-stained dense collagen.

IHC staining of PERK was performed to evaluate the effect of GSK2606414 on the inhibition of the PERK signaling pathway (Figure 7b). In the sham group, no positive PERK signal expression in the periodontium was observed as a consequence of the lack of inflammation. In the model group that had periodontitis induced with LPS for three weeks, the expression of PERK increased significantly. The expression levels of these cytokines in the LPS + Free GSK group and the NGHM hydrogel group were noticeably reduced compared with those in the LPS group. Furthermore, GSK2606414 partially blocked the LPS-induced alveolar bone defects in rats by inhibiting the expression of PERK pathway signaling molecules in the UPR and inhibiting the occurrence of the endoplasmic reticulum stress response. The area of blue collagen fibers and the mean fluorescence intensity of PERK in the immunohistochemical staining map were measured using ImageJ 1.53p software, and the semiquantitative data (Figure 7c,d) were consistent with the staining map results. These results indicate that GSK2606414-loaded hydrogels could conspicuously inhibit the inflammation of the periodontal tissue, which could greatly reduce the destruction of the periodontium.

## 4. Conclusions

In summary, we successfully prepared a collagenase-responsive drug-loaded hydrogel system and applied it for periodontitis treatment. PDLLA-PEG-PDLLA was successfully synthesized using a ring-opening reaction, and NanoGSK was prepared by using PDLLA-PEG-PDLLA as an excipient in combination with the HGNPT method. In addition, an enzyme-sensitive drug-loaded hydrogel precursor material (HA-AC) with MMPc as the cross-linking agent was successfully prepared. By studying the degradation and in vitro drug release behavior of NGHM, it was demonstrated that it had obvious enzyme responsiveness to achieve controlled and slow drug release. In the cellular uptake, cytotoxicity, and cell proliferation assays, NanoGSK was demonstrated to be taken up by cells without affecting the growth and proliferation of normal cells. The in vitro osteogenic ability and in situ regeneration of periodontal tissues were performed under simulated periodontitis conditions. The experimental results showed that the drug-loaded hydrogels had good responsiveness to controlled and slow release. Therefore, we believe that our work provides a promising topical stimulus-responsive drug delivery system with broad application prospects for the treatment of periodontitis.

## Data Availability

All data are contained in the article.

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
