# Peer review of "Collagenase-Responsive Hydrogel Loaded with GSK2606414 Nanoparticles for Periodontitis Treatment through Inhibiting Inflammation-Induced Expression of PERK of Periodontal Ligament Stem Cells"

_pharmaceutics, 2023, doi:10.3390/pharmaceutics15102503_

Round 1

Reviewer 1 Report

The authors developed a collagenase-responsive hydrogel loaded with GSK2606414 and investigated its impact on periodontitis. The comments are as follows:

1.The application of the MMP-2 cleaved peptide cross-linker (MMPc) to generate hydrogels has been discussed. Given that MMP levels increase in periodontitis, it is crucial to assess whether MMPc could influence the progression of periodontitis.

2.The resolution presented in Figure 2 is a bit of low. It is advisable to make the necessary adjustments. Additionally, please provide quantifications for the cell update rate.

3.In Figure 2b, it is mentioned that hydrophobic drugs impact cell survival in cell culture. However, there seems to be a contradiction between line 394 and line 395 regarding the cytotoxicity of NanoGSK and NGHM. Additionally, it is suggested to employ software for statistical analysis of differences between groups in Figure 2c.

4.The "2.11. Periodontal Tissue Regeneration in Situ" section states the use of twenty-five six-week-old Sprague-Dawley rats, divided into five groups with six rats per group. This would make a total of 30 rats, not 25.

5.It is recommended to review Figure 5 and clearly label the group that corresponds to LPS+nasoGSK. Quantifying bone loss across various groups is essential for assessing the impact of hydration on the progression of periodontitis.

6.The assertion that GSK2606414-loaded hydrogels noticeably inhibit periodontal tissue inflammation (line 537) is addressed. It is noted that inflammation cytokines were not assessed across different groups, and histological staining alone (H&E staining) may not suffice to determine inflammation status accurately.

7.There's a discrepancy between the mentioned division of rats into six groups for "Periodontal Tissue Regeneration in Situ" in the methods section and the presence of a "Sham" group in Figures 6 and 7, but no data for the LPS+NanoGSK group. This discrepancy should be clarified.

8.The concern is raised that although the manuscript mentions the involvement of inflammation cytokines and ER stress in the effects of biomaterials on periodontitis, no detection of these factors is presented. It is suggested that the logical connection between these claims and the lack of supporting data should be addressed.

Moderate editing of English language required

Author Response

  1. The application of the MMP-2 cleaved peptide cross-linker (MMPc) to generate hydrogels has been discussed. Given that MMP levels increase in periodontitis, it is crucial to assess whether MMPc could influence the progression of periodontitis.

Response: Thanks very much for the reviewer’s comments. The MMP-2 cleaved peptide cross-linker (MMPc) containing amino acid sequences of GCRDGPQGIWGQDRCG was widely used as a cross-linker to fabricate hydrogel scaffold for 3D stem cell culture (Materials Science & Engineering C 108 (2020) 110276; Biomaterials 31 (2010) 7836-7845), there is no evidence to suggest that MMPc has negative impact on cell growth and tissue inflammation. Therefore, in our work, we didn’t investigate the influence of MMPc on the progression of periodontitis.

  1. The resolution presented in Figure 2 is a bit of low. It is advisable to make the necessary adjustments. Additionally, please provide quantifications for the cell update rate.

Response: Thanks very much for the reviewer’s comments. We believe that the issue raised by the reviewer refers to Figure 3 instead of Figure 2. We have adjusted Figure 3 to increase the resolution of the image while maintaining authenticity.

We have added quantification data of the cell update rate as Figure 3b.

  1. In Figure 2b, it is mentioned that hydrophobic drugs impact cell survival in cell culture. However, there seems to be a contradiction between line 394 and line 395 regarding the cytotoxicity of NanoGSK and NGHM. Additionally, it is suggested to employ software for statistical analysis of differences between groups in Figure 2c.

Response: Thanks for this comment. We believe that the issue raised by the reviewer refers to Figure 3 instead of Figure 2. We have corrected these incorrect explanations about Figure 3c in the revised version of the manuscript, and added more appropriate explanations (Page 10, line 413-417).

 In addition, we have revised Figure 3(d) and 3(e) with statistical analysis.

  1. The "2.11. Periodontal Tissue Regeneration in Situ" section states the use of twenty-five six-week-old Sprague-Dawley rats, divided into five groups with six rats per group. This would make a total of 30 rats, not 25.

Response: Thank you very much for the detailed instructions. We have revised the manuscript (Page7, line 278).

  1. It is recommended to review Figure 5 and clearly label the group that corresponds to LPS+nasoGSK. Quantifying bone loss across various groups is essential for assessing the impact of hydration on the progression of periodontitis.

Response: Thanks very much for the reviewer’s comments. We have modified the Figure 5 with clearer labels. For the in vivo experiments, we have used SD rat models to demonstrate the prominent therapeutic effects of NanoGSK-loaded hydrogels as compared to untreated group (LPS group) and FreeGSK treated group, so that there is no LPS+NanoGSK treated group was investigated in vivo.

We have added a statistical graph for the distance of CEJ-ABC as Figure 5f, and the explanations was added in the revised manuscript (Page13, line 518-522 ).

  1. The assertion that GSK2606414-loaded hydrogels noticeably inhibit periodontal tissue inflammation (line 537) is addressed. It is noted that inflammation cytokines were not assessed across different groups, and histological staining alone (H&E staining) may not suffice to determine inflammation status accurately.

Response: Thanks for this comment. The destruction of the alveolar bone and gingival area could be observed by H&E staining, but the amount of expression of inflammatory cytokines was the focus of our follow-up work.

  1. There's a discrepancy between the mentioned division of rats into six groups for "Periodontal Tissue Regeneration in Situ" in the methods section and the presence of a "Sham" group in Figures 6 and 7, but no data for the LPS+NanoGSK group. This discrepancy should be clarified.

Response: Thanks for this comment. In this work, NanoGSK was prepared to allow better dispersion of GSK in the hydrogel to form a homogeneous system for controlled drug release behavior. Therefore, for in vivo experiment design, our main goad is to compare the thereapetic efficacy of FreeGSK and drug-loaded hydrogels, and  the NanoGSK group was not studied. We have corrected the errors of presentation in the manuscript (Page 7, line 292).

  1. The concern is raised that although the manuscript mentions the involvement of inflammation cytokines and ER stress in the effects of biomaterials on periodontitis, no detection of these factors is presented. It is suggested that the logical connection between these claims and the lack of supporting data should be addressed.

Response: Thanks very much for the reviewer’s comments. Based on previous report, GSK2606414 is a novel and highly selective PERK inhibitor, and it effects by binding to the active site of PERK, inhibiting the activation of phosphorylation of eukaryotic transcription initiation factor 2α (elF2α) downstream of PERK, and regulating the onset of unfolded protein response (UPR) due to ER stress (Sci. Transl. Med. 2013, 5.; Cell Cycle 2014, 13, 801-806.; ACS Med. Chem. Lett. 2013, 4, 964-968.).  In our futher works, we will study in detail for GSK2606414 regulating the expression of PERK, TNF-α, and IL-1β cytokines during the treatment of the periodontitis.

Comments on the Quality of English Language

Moderate editing of English language required

Response: Thanks for this comment. We have carefully polished the English writing of the manuscript.

Reviewer 2 Report

The current manuscript submitted to Pharmaceutics exhibits studies regarding the synthesis of collagenase responsive hydrogels loaded with nanodrugs for periodontitis treatment. In vitro and in vivo tests showed good results for the developed nanomaterials.

The manuscript is very well written, the studies are clearly and in detail presented, and the results are of interest for stomatological research.

I recommend the manuscript to be published with some minor observations:

-          To add the company details in the Materials chapter -company, city, country.

-          To label also the columns in Figure 5, and modify/complete the caption.

I have also noticed that an information regarding the following should be added:

Institutional Review Board Statement: In this section, please add the Institutional Review Board Statement and approval number for studies involving humans or animals. Please note that the Editorial Office might ask you for further information. Please add “The study was conducted according to the guidelines of the Declaration of Helsinki, and approved by the Institutional Review Board (or Ethics Committee) of NAME OF INSTITUTE (protocol code XXX and date of approval).” OR “Ethical review and approval were waived for this study, due to REASON (please provide a detailed justification).” OR “Not applicable” for studies not involving humans or animals. You might also choose to exclude this statement if the study did not involve humans or animals.

Author Response

I recommend the manuscript to be published with some minor observations:

-          To add the company details in the Materials chapter -company, city, country.

Response: Thanks for your comment. We have already added more detailed information of the reagents in the Section 2.1.

-          To label also the columns in Figure 5, and modify/complete the caption.

Response: Thanks for pointing out this mistake. We have modified the Figure 5 and added the labels of different columns in the Figure.

I have also noticed that an information regarding the following should be added:

Institutional Review Board Statement: In this section, please add the Institutional Review Board Statement and approval number for studies involving humans or animals. Please note that the Editorial Office might ask you for further information. Please add “The study was conducted according to the guidelines of the Declaration of Helsinki, and approved by the Institutional Review Board (or Ethics Committee) of NAME OF INSTITUTE (protocol code XXX and date of approval).” OR “Ethical review and approval were waived for this study, due to REASON (please provide a detailed justification).” OR “Not applicable” for studies not involving humans or animals. You might also choose to exclude this statement if the study did not involve humans or animals.

Response: We have added relevant content in the revised manuscript (line 280-283). The revised parts were highlighted in red color.

Reviewer 3 Report

The authors used an interesting approach involving collagenase responsive hydrogel loaded with PERK inhibitor nanodrug for treatment of periodontitis by inhibiting inflammation in periodontal ligament stem cells.

Comments

1.      Lines 35-39; 48-49; 551-554: These sentences are not clear. They should be rephrased.

2.      Line 93: Reference is required at the end of this sentence.

3.      Line 235: The authors should indicate the “above method”.

4.      Methods:

-          The number of animals in each group should be indicated

-          Methods used for qRT-PCR gene expression, primer sequences, histological treatments, and Western blot analyses are missing. They should be described.

5.      Lines 363-374: References should be supplied after each statement.

6.      Fig 4D,E: In qRT-PCR Control sample should be equal to 1. The graphs should be corrected.

7.      Fig 4D,E: The term Relative Intensity should be defined.

8.      Scale bars are missing on Figs 6,7. This should be corrected.

Author Response

  1. Lines 35-39; 48-49; 551-554: These sentences are not clear. They should be rephrased.

Response: Thanks for your comment. We have modified the sentence and highlighted in red color.

  1. Line 93: Reference is required at the end of this sentence.

Response: Thanks for this comment. We have inserted relevant references.

  1. Line 235: The authors should indicate the “above method”.

Response: Thanks for pointing out this mistake. We have modified the sentence and highlighted on line 235.

  1. Methods:

- The number of animals in each group should be indicated

- Methods used for qRT-PCR gene expression, primer sequences, histological treatments, and Western blot analyses are missing. They should be described.

Response: Thanks for this comment. We have indicated the number of animals in each group in the manuscript at line 283. In addition, we have added relevant experimental methods in the revised manuscript and highlighted in red color.

Table 1 Primers for qRT-PCR

Gene

Prime Sequence (5’~3’) F: forward R: reverse

Runx2

F: CCACCACTCACTACCACACCTAC

R: CCTCACACTCCTCGCCCTATTG

OCN

F: CCTCACACTCCTCGCCCTATTG

R: TCAGCCAACTCGTCACAGTCC

GAPDH

F: GAGAAGGCTGGGGCTCATTT

R: AGTGATGGCATGGACTGTGG

  1. Lines 363-374: References should be supplied after each statement.

Response: Thanks very much for the reviewer’s comments. We have inserted relevant references.

  1. Fig 4D,E: In qRT-PCR Control sample should be equal to 1. The graphs should be corrected.

Response: Thanks for this comment. We chosed the LPS group as a control sample to calculate the relative expression of mRNA in each group. So the Control group was not equal to 1.

  1. Fig 4D,E: The term Relative Intensity should be defined.

Response: Relative Intensity indicated the gene expression of each group using the LPS group as a control, and values were calculated according to the 2-∆∆ Ct method. We have increased it in the revised manuscript and highlighted in red color.

  1. Scale bars are missing on Figs 6,7. This should be corrected.

Response: Thanks for pointing out this mistake. We have corrected Figure 6 and 7 in the revised manuscript.

Comments on the Quality of English Language

Lines 35-39; 48-49; 551-554: These sentences are not clear. They should be rephrased.

Response:Thank you very much for the highly positive and encouraging comments. We have added more discussions about the nanoformulation aspects as well as other results. In addition, we have also revised the introduction and the conclusion parts to improve the scientific insight to the readers.